# A Tangled Threesome: Circadian Rhythm, Body Temperature Variations, and the Immune System

**DOI:** 10.3390/biology10010065

**Published:** 2021-01-18

**Authors:** Benjamin Coiffard, Aïssatou Bailo Diallo, Soraya Mezouar, Marc Leone, Jean-Louis Mege

**Affiliations:** 1Microbes Evolution Phylogénie et Infection, Institut Recherche et Développement, Aix-Marseille University, 13005 Marseille, France; ayebailo66@gmail.com (A.B.D.); soraya.mezouar@univ-amu.fr (S.M.); marc.leone@ap-hm.fr (M.L.); 2Institut Hospitalo-Universitaire-Méditerranée Infection, Assistance Publique-Hôpitaux de Marseille, 13005 Marseille, France; 3Médecine Intensive-Réanimation, Hôpital Nord, Assistance Publique-Hôpitaux de Marseille, 13915 Marseille, France; 4Service d’Anesthésie et de Réanimation, Hôpital Nord, Assistance Publique-Hôpitaux de Marseille, 13915 Marseille, France

**Keywords:** body temperature, circadian rhythm, chronobiology disorders, immunity, disease

## Abstract

**Simple Summary:**

In mammals, including humans, the body temperature displays a circadian rhythm and is maintained within a narrow range to facilitate the optimal functioning of physiological processes. Body temperature increases during the daytime and decreases during the nighttime thus influencing the expression of the molecular clock and the clock-control genes such as immune genes. An increase in body temperature (daytime, or fever) also prepares the organism to fight aggression by promoting the activation, function, and delivery of immune cells. Many factors may affect body temperature level and rhythm, including environment, age, hormones, or treatment. The disruption of the body temperature is associated with many kinds of diseases and their severity, thus supporting the assumed association between body temperature rhythm and immune functions. Recent studies using complex analysis suggest that circadian rhythm may change in all aspects (level, period, amplitude) and may be predictive of good or poor outcomes. The monitoring of body temperature is an easy tool to predict outcomes and maybe guide future studies in chronotherapy.

**Abstract:**

The circadian rhythm of the body temperature (CRBT) is a marker of the central biological clock that results from multiple complex biological processes. In mammals, including humans, the body temperature displays a strict circadian rhythm and has to be maintained within a narrow range to allow optimal physiological functions. There is nowadays growing evidence on the role of the temperature circadian rhythm on the expression of the molecular clock. The CRBT likely participates in the phase coordination of circadian timekeepers in peripheral tissues, thus guaranteeing the proper functioning of the immune system. The disruption of the CRBT, such as fever, has been repeatedly described in diseases and likely reflects a physiological process to activate the molecular clock and trigger the immune response. On the other hand, temperature circadian disruption has also been described as associated with disease severity and thus may mirror or contribute to immune dysfunction. The present review aims to characterize the potential implication of the temperature circadian rhythm on the immune response, from molecular pathways to diseases. The origin of CRBT and physiological changes in body temperature will be mentioned. We further review the immune biological effects of temperature rhythmicity in hosts, vectors, and pathogens. Finally, we discuss the relationship between circadian disruption of the body temperature and diseases and highlight the emerging evidence that CRBT monitoring would be an easy tool to predict outcomes and guide future studies in chronotherapy.

## 1. Introduction

In mammals, including humans, the body temperature displays a circadian rhythm and is maintained within a narrow range to facilitate the optimal functioning of physiological processes [1]. Physiological body temperature is approximately 37.0 °C in healthy subjects, with a nearly 1 °C sinusoidal circadian fluctuation [1,2]. The core body temperature is the result of a fine balance between heat production and heat loss, with the lowest level of temperature arriving at the resting phase during the night and the highest level at the end of the day as a consequence of the physiological metabolism (Figure 1). This process is due to several mechanisms, including (1) conversion of chemical energy from foods to heat metabolic [3] and mechanical energy from muscular contraction (physical efforts, shivering) [4], (2) cellular oxidative metabolism that produces a constant and stable source of heat, and (3) dissipation of heat through sweating or vasomotor changes that regulate blood flow to the skin and mucous membranes [5]. Heat is lost at the skin surface by the mechanisms of convection, radiation, and evaporation or is dissipated by the respiratory system (breathing) [6].

Body temperature in mammals is under the control of the biological clock [1]. The master clock that controls all biological rhythm resides in the suprachiasmatic nucleus (SCN) of the brain [7,8]. The biological clock is cell-autonomous and in its simplest form consists of a transcription–translation oscillator loop. In the SCN and most somatic cells, the core molecular clock is generated by the heterodimer BMAL1:CLOCK proteins that drive the expression of their own repressors (mainly PER and CRY proteins) [9]. The transcription factors within each of these loops can also regulate clock-controlled genes (*Ccg*). The latter express a circadian profile and regulate numerous functions including immunity [9,10].

A close relationship is now established between body temperature, the biological clock, and immune functions and diseases [10,11]. The present review aims to characterize the circadian rhythm of the body temperature (CRBT) in humans and its relation to immunity from molecular pathways to diseases. The origin and homeostatic mechanisms of CRBT will be briefly mentioned. Reasons for body temperature disruption; immune and biological effects of temperature variability in the host, vector, and pathogen; and the association between circadian disruption of the body temperature and diseases will be described.

## 2. The Circadian Rhythm of the Body Temperature

### 2.1. Clock Control of the CRBT

Internal fluctuations of the body temperature are regulated by the circadian rhythm. The latter is under the control of the SCN located in the hypothalamus, which is also the dominant thermoregulatory controller in mammals [12,13]. Temperature is sensed by the transient receptor potential (TRP) family of ion channels [14]. Thermal TRPs are activated at distinct temperature thresholds and are typically expressed in sensory neurons. The subtype TRPV3 senses heat [15], whereas cold is sensed by TRPM8 [16]. Thus, thermal information from the skin surface, peripheral tissues, core organs, and neuraxis per se are integrated at various levels, finally arriving at the hypothalamus. The direct control of the SCN on the hypothalamic thermoregulatory center was never proven in humans but highly supposed. In a model of squirrels, the CRBT was permanently prevented in animals with sustained complete ablation of the SCN [12]. The CRBT is most likely synchronized by the rhythmic input from the SCN acting upon the hypothalamic thermoregulatory centers. These centers modulate the setpoint and alter the thresholds for cutaneous vasodilatation and sweating.

More recently, Nam et al. [17] showed that the formation and metabolic functions of the brown adipose tissue, a key organ for body temperature maintenance, are controlled under an orchestrated circadian clock regulation. They observed the cell-intrinsic clock machinery exerts concerted control of brown adipogenesis with consequent impacts on adaptive thermogenesis. These data highlight a thermogenic capacity of fine-tuning brown adipose tissue that complements the temporal mechanisms of circadian regulation of body temperature.

### 2.2. Measurements of the CRBT

The clock control of body temperature includes a circadian cycle consisting of an amplitude (nearly 1 °C of difference from min to max values) and a period (24 h for a complete cycle) [18]. Figure 2 depicts all parameters and the nomenclature of the CRBT. The method of temperature monitoring is a critical issue in the study of CRBT. Temperature can be monitored at different sites, with the choice of the site resulting in certain trade-offs in terms of convenience and reliability. Peripheral measurements (oral, axillary, or thoracic skin surface), which are generally more convenient, have a comparable amplitude of temperature variation than central measurement (rectal or intestinal). However, most likely due to the heat radiation from central to peripheral tissues, peripheral methods of monitoring have a far higher variance and range of periods of time [18,19]. Thus, central monitoring appears to be the most accurate monitoring to assess the CRBT [20].

### 2.3. Physiological Modulations of the CRBT

Many environmental and physiological factors have been reported to influence the CRBT in humans. The change of the CRBT may have consequences on physiological functions (see below). This section aims to describe the physiological change in CRBT in some specific situations (i.e., mainly seasons, age, and sex), as CRBT is known to be associated with different immune response profiles and infectious risks [21,22]. The role of physiological variation in temperature circadian rhythm in the immune response was never assessed, but it could be an interesting field of research in the future.

Several studies revealed that the time of year (seasons) might be a possible source of body temperature variability [18]. A large-scale study (*n* = 93,225) demonstrated robust and consistent behavior of the human circadian cycle at the population level [23]. They showed that over the year, body temperatures were slightly colder in winter than summer (~0.2 °C difference) (Figure 1). They suggested that seasonal variation of temperature might be due to ambient effects on body temperature that are not eliminated because they fall within the tolerance range of the thermoregulatory system. In addition, the bathyphase (daily time of minimum temperature) appeared to parallel sunrise times, and the acrophase (daily time of maximum temperature) and sunset times followed opposite seasonal patterns, with acrophase preceding nightfall in summer and following nightfall in winter.

The circadian system undergoes dramatic changes during an individual’s lifetime, particularly during early ontogenetic development and in old age [24]. There is some evidence that daily temperature level and amplitude of temperature decrease with aging [25,26]. It was suggested that these changes may be related to less efficient intrinsic mechanisms of thermoregulation but could also be linked to a sedentary lifestyle, chronic diseases, or medications, and may affect other circadian functions [24].

The CRBT is different according to gender and mainly influenced by the sexual hormones [27]. In females, the body temperatures vary in a predictable manner across the menstrual cycle in a normally cycling individual. An increase in body temperature ranging from 0.25–0.5 °C is typically observed during the ovulation period. Although females maintain a similarly shaped circadian body temperature curve in both the follicular and luteal phases, a decrease in amplitude occurs. Indeed, the increase in temperature value at bathyphase with ovulation is not accompanied by the same degree of increase in peak temperature values [28]. The intrinsic circadian period of the body temperature was shown to be significantly shorter in women but only in young subjects highlighting the potential mixed effect of gender and age [29].

Changes in the circadian rhythm of body temperature can also occur during a disruptive environment, illness, or medication. A large number of studies have also shown a close association between the CRBT and sleep disorders [30]. Night workers display significant circadian rhythm abnormalities, including disruption of the body temperature that persists even after retirement [31,32].

## 3. Molecular, Immune, and Biological Effects of Temperature Variations

### 3.1. Transcriptional Effect of Temprature Variations

Temperature cycles have been shown to function as systemic indices that effectively drive the phase of individual oscillators in cultured cells and tissue explants (Figure 3A) [9,11,33]. Simulated body temperature cycles of mice and even humans, with daily temperature differences of only 3 °C and 1 °C, respectively, but also considerably longer or shorter periods than 24 h, could gradually synchronize the expression of circadian genes in cultured mouse fibroblasts [11]. In Drosophila, temperature cycles not only induce oscillations of clock proteins but also synchronize behavioral rhythms, revealing an effect on physiological functions. The temperature-induced rhythms were also observed under constant light conditions, a situation that normally leads to molecular and behavioral arrhythmicity, revealing a tissue-autonomous process that can override the effects of light [34]. In a mouse model, temperature drive resistance has been demonstrated as a property of the SCN network and not an autonomous cellular property of mammalian clocks. This differential sensitivity to temperature allows the SCN to drive circadian rhythms in body temperature, which can then act as a universal cue for the entrainment of cell-autonomous oscillators throughout the body [35].

The molecular mechanism involving the temperature in the clock transcriptional loop is depicted in Figure 3B. The transcriptional effect of the temperature rhythm involves the cold-inducible RNA-binding protein (CIRP), a highly conserved RNA-binding nuclear protein that is upregulated at lower temperatures. CIRP modulates circadian gene expression post-transcriptionally and appears to be a widespread feature in the temperature-dependent regulation of mammalian gene expression [37]. Heat is also involved in the mechanism. Heat shock factor 1 (HSF1) is a circadian transcription factor that binds heat shock element (HSE) sequence in a daily rhythmic manner, leading to the circadian activation of HSF1 target promoters, including *Per2* [38]. Thus, a heat shock at 40 °C for 150 min (simulating a fever) in the cultured liver and lung explants of mice induced a strong increase of *mPer2* expression [39], confirming the close link between CRBT and circadian rhythm.

Inversely, the molecular clock is crucial for generating circadian rhythms, including body temperature. In a mouse model, the non-coding cis-element of *Per2*, one of the main clock genes, has been demonstrated to be essential for maintaining body temperature rhythmicity [40].

As explained above, the transcription factors generated by the molecular clock regulate clock-control genes (*Ccg*) [9]. In several animal models, there is strong evidence that many immune genes are under the control of the molecular clock [10]. In humans, mistimed sleep affects the molecular regulators of circadian rhythmicity and leads to a reduction of rhythmic transcripts in the human blood transcriptome from 6.4% at baseline to 1.0% during forced desynchrony of sleep [41]. Many of the circadian dysregulated genes are involved in immune pathways [42]. To our knowledge, the direct link between the CRBT and the expression of the immune genes was not specifically shown; however, by extrapolation, one can consider that disruption of the CRBT might most likely affect the expression of the *Ccg*, including immune genes.

### 3.2. Effect of Temperature Change on Immune Function

Many variables of the human immune system exhibit distinct 24-h rhythms, such as the number of circulating leukocytes or levels of pro- and anti-inflammatory cytokines [43]. Thus, circadian disruption may have a negative impact on these features. Numerous studies have addressed the influence of sleep–wake cycles on the circadian rhythm of the immune actors [44]. The SCN most likely conveys timing information to the immune system mainly through autonomic and endocrine pathways involving cortisol and melatonin, or through temperature variations. These signals promote phase coherence of peripheral clocks in the immune system and also govern daily variations in immune function [10].

The effect of the CRBT on the immune effectors has never been directly assessed in humans. However, the level of body temperature and its rhythm can have varying impacts on immunity (Table 1) [42,45,46,47,48,49,50,51,52,53,54,55,56,57,58,59,60,61,62,63]. Elevated body temperatures generally promote the activation, function, and delivery of immune cells, whereas reduced temperatures inhibit these processes [64]. As detailed above, a change of temperature influences the molecular clock expression. Thus, an elevation of body temperature above 38 °C (fever) is likely a trigger to activate circadian metabolic pathways, such as immune functions. Conversely, hypothermia (<36 °C) has been associated with a poor prognosis in critically ill patients [65]. In animal models, hypothermia affects lymphocyte recirculation patterns and trafficking molecule expression, primarily by exerting an anti-inflammatory influence [63]. At a molecular level, hypothermia may lead to an accumulation of the CIRP protein and disrupt the expression of the molecular clock [36].

On the scale of a day (or of the circadian rhythm), one hypothesis would be that an increase in temperature (which takes place during the day) prepares the organism to fight aggression specifically in the daytime (it is more probable that trauma or infection would occur during the day rather than the night). It has been demonstrated that an infection is more severe when mice are infected at the resting phase (low level of temperature) than during the active phase (high level of temperature) [66].

The effects of temperature variability on immune function have been only assessed in a few studies conducted on plants and animal models, which suggest an ancestral molecular mechanism. In a similar manner to mammals, plants present a gene circuit with negative feedback to maintain and regulate the clock over a 24-h period. Moreover, similar to mammals, their circadian clock can be entrained by the environment such as light, nutrients, or temperature [67]. Plants are exposed to their environment and significant changes in ambient air temperature according to the day/night and seasons. Change in climatic temperature is often associated with changes in other abiotic factors, such as light and humidity, but there is evidence for temperature-mediated modulation of defense responses. The molecular effect in plants of temperature involves a group of proteins encoded by the R genes. The temperature R-mediated response is involved in the modulation of host resistance against viral and fungal pathogens in different models of plants [68]. Besides, ectotherm animals are organisms that have internal physiological sources of heat relatively small or quite negligible importance in controlling body temperature. Such organisms (for example, frogs) rely on environmental heat sources and are very sensitive to rapid changes in air temperature. In an amphibian model, animals exposed to repeated cycles (of 24 h period) of temperature fluctuations (1–3 weeks) had a 70% greater plasma activity against *Pseudomonas aeruginosa* and a 50% increase in *Escherichia coli*-killing capacity compared with the control group maintained at a constant temperature [69]. In this model, the levels of circulating immune cells were not affected by temperature fluctuations, suggesting a molecular qualitative effect rather than a quantitative change in immune parameters. Crayfish, another ectothermic animal, exposed to temperature cycles (of 24-h periods) not only exhibited a significant increase in resistance to *Aeromonas hydrophila* but also a circadian variation of hemocyte count compared with animals maintained at a constant temperature [70]. For mice exposed to 12:12-h light:dark schedule, T-lymphocytes showed a strong association with body temperature rhythm, suggesting a likely temperature control on lymphocyte function [71]. The influence of body temperature level on immune functions has been so far well studied in humans, but it would be interesting in future studies to focus specifically on the effect of temperature variability (or disruption of normal CRBT) and its effects on the molecular clock and the immune response rather than simply analyzing a temperature level.

### 3.3. Effect of Temperature Variations on Microbes

It is now well-established that microorganisms, such as parasites or bacteria, express circadian rhythm [72,73]. Cyanobacteria are a group of photosynthetic bacteria in which circadian rhythms are endogenously generated by a unique KaiABC protein clock [74]. The KaiABC complex and the circadian rhythm it generates is entrained to the ambient light:dark cycle through photosynthetic changes in the ATP/ADP ratios [75]. The universal analysis of BLAST showed that this circadian clock was present less abundantly in other bacteria and archaea, and responsibility for circadian behavior remains uncertain, especially in non-photosynthetic bacteria [76]. Besides, it has been demonstrated that temperature variability may entrain circadian rhythm in microbes. *Trypanosoma brucei*, an extracellular parasite (the causative agent of human sleeping sickness) has circadian transcriptomic oscillations that are entrained by temperature cycles. Almost all of the temperature-entrained transcript oscillations (96%) were absent under light:dark cycle conditions, suggesting a non-autonomous and non-photonic entrain mechanism [77]. Recently, the enteric proteobacterium *Klebsiella aerogenes*, a human gastrointestinal commensal, has been shown to express an endogenously generated circadian rhythm that can be entrained to changes in ambient temperature in a model of bioluminescent culture [78]. Bacteria were entrained by different periods and amplitudes of temperature with stable phase relationships. The authors hypothesized that the *Klebsiella aerogenes* circadian clock entrains to its host via detection of and phase shifting to the daily pattern of the CRBT.

Recently, evidence has emerged revealing oscillations of fecal microbiota during the 24-h cycle—the gut microbiome exhibits compositional and functional structures at different times of day [79,80]. Although exposed to environmental changes, such as nutrient availability [81] and the level of host-derived immunity [80], the CRBT could be considered an important circadian parameter influencing the gut microbiome.

### 3.4. Effect of Temperature Variations on Vectors

Ecological research has revealed that environmental factors, such as fluctuating temperatures, can strongly affect insect and invertebrate’s immunity [82]. Butterflies (*Lycaena tityrus*) exposed to temperatures fluctuating around cooler (17.7 °C) and warmer (23.7 °C) than normal temperatures experience significantly higher phenoloxidase activity and higher total hemocyte numbers compared with butterflies housed at a constant temperature [83]. The overall effect of temperature on the ability of a vector to resist infection depends on the effects on elements of vector immune function and physiology; it might also be related to direct effects of temperature on the microbe itself as exposed above.

## 4. Circadian Disruption of the Temperature and Diseases

### 4.1. Fever and Hypothermia

Pyrexia (also named fever) is the altering upward of the thermoregulatory set point, often secondary to the systemic inflammatory response to a stimulus, such as infection. As explained above the elevation of body temperature generally promotes a beneficial response, such as the activation, function, and traffic of immune cells [65]. In septic patients subsequently admitted to ICU, fever is associated with lower mortality and shorter hospital stays, suggesting the beneficial effect of an increase of body temperature in infection [84]. Inversely, a high peak of body temperature in critically ill patients without brain injury is also an independent predictor of mortality [85]. A high level of body temperature could indeed be a marker of an excessive inflammatory response and a rupture of immune homeostasis that can be harmful to the patient [86]. Fever may also be a predictor of poor outcome in other non-septic diseases, such as brain injury patients, acute pulmonary embolism, cardiogenic shock, or the postoperative phase of gastric cancers [87,88,89,90,91].

Conversely, hypothermia is often associated with a less effective immune response and a worse prognosis than fever [64,66]. Moreover, in the last few decades, prophylactic hypothermia has been suggested to be neuroprotective in many neurological diseases by, in particular, a reduction of the inflammatory response. However, recent literature did not demonstrate any effect of induction of hypothermia as compared with normothermia, and the potential benefit of such a strategy is most likely related to the prevention of fever [92,93].

Change of temperature level is often associated with diseases, which confirms the clinical link between temperature and host response.

### 4.2. Disruption of the CRBT and Trauma

Acute trauma or burn patients are a specific subgroup of the critically ill population due to sudden injury and dramatic changes in homeostasis producing a high systemic inflammatory response, which may result in a change of the temperature course and be associated with fever [94]. Moreover, severe trauma provokes a stress-induced activation of the hypothalamic–pituitary–adrenocortical axis [95]. Significant circadian rhythm alterations of melatonin and cortisol were reported in this population most likely as the consequence of the systemic response and reported as a marker of severity [96,97]. Disruption of the CRBT was reported in severe trauma as well. Two studies, including ours, assessed continuous body temperature measurement in severe trauma patients. Body temperature is easier to collect than biomarkers and allows continuous data collection at a high frequency. Thus, precise circadian rhythm variables including the frequency of oscillation (period) and the amplitude may be easily assessed by mathematical modeling. In severely brain-injured patients, Blume et al. provided evidence for an association between periods of temperature closer to 24-h and a good neurological outcome [98]. Our study on severe trauma showed a correlation between lower body temperature and mortality and, interestingly, a correlation between higher amplitude in body temperature and mortality [99]. This finding correlates with a study suggesting a poor prognosis in the patients experiencing high peaks of body temperature [85] that could be a marker of excessive circadian activation and non-adapted systemic inflammatory response. Interestingly, this result was mainly found in patients with traumatic brain injury. Consequently, we hypothesized that traumatic brain injury could be a more direct way of molecular clock disruption in the SCN as already shown in a rat model [99,100].

### 4.3. Disruption of the CRBT and Infection

It is nowadays well-established that infections are closely related to circadian rhythms in terms of susceptibility, clinical presentation, and severity [101]. Interactions between circadian and immune systems are bidirectional, such that immune factors can influence circadian timing by acting on the biological clock in the SCN and clocks in peripheral tissues [9,10]. The relation between sepsis and circadian disruption in humans was particularly well-described in ICU patients [102]. The first study of body temperature rhythms in 15 ICU patients found that despite significant rhythm in a period of 24-h in 80% of the patients, the position of acrophase (peak time) of the rhythm varied markedly both between patients and within patients. They observed also a tendency for the amplitude of the temperature rhythm to be greater when the patients were unconscious and in those who did not survive the ICU stay. [103]. These findings are in line with our study in severe trauma patients [99] and the hypothesis that high amplitude could be a marker of severity and probably excessive circadian activation. Gazendam et al. showed that 17 of 21 ICU patients had the circadian phase position (peak time) that fell earlier or later than the normative range (i.e., around 6 pm) [104]. In another study in ICU patients, based on recurrence quantification analysis, a method of nonlinear data analysis (chaos theory) for the investigation of dynamical systems, authors found a decrease in temperature amplitude below normal range at ICU admission in septic patients and lower severity of disease and better clinical outcomes in patients with higher temperature amplitude [105]. These data suggest that the circadian rhythms may be a good mechanism of adaptation when correctly solicited.

The CRBT was less studied in non-ICU infectious diseases in humans. Sothern et al. reported that temperature acrophases (peak) obtained by monitoring oral temperature in individuals infected with HIV were more variable than those of healthy subjects [106]. In animals, some studies found an association between disruption of the CRBT and infectious diseases such as in African cattle infected with trypanosomes and tick-borne infections, or monkeys infected with simian immunodeficiency virus. The measure of rectal temperature, which is commonly used to assess pyrexia during clinical diagnosis of bovine Indigenous Nkedi Zebu cattle, was found increased only in the afternoon [107]. Moreover, in monkeys infected with simian immunodeficiency virus, authors found a progressive decrease in the temperature amplitude and a significant delay of the acrophase compared with non-infected monkeys [108].

### 4.4. Disruption of the CRBT and Cancer

In cancer patients, some old studies monitored oral or skin temperatures in patients with advanced cancer. Individual analyses showed large differences in the circadian rhythm of temperature among subjects with poor performance, highlighting the association between circadian disruption and disease severity [109,110,111].

Chemotherapy has been also demonstrated to be associated with circadian disruption of body temperature [112,113]. Circadian timing of anti-cancer medications has been shown to improve treatment tolerability up to fivefold and double efficacy in experimental and clinical studies [114] and highlights the potential role of temperature monitoring in personalized chronotherapeutic [115].

### 4.5. Disruption of the CRBT and Inflammatory Diseases

Inflammatory diseases are subject to systemic inflammation, and the importance of circadian rhythm in the pathophysiology of asthma, inflammatory arthritis, or inflammatory bowel disease has been reported many times [116,117,118]. One study in psoriasis patients assessed body temperature and showed variable acrophases (peak time) and desynchronization with other markers of circadian rhythm compared with control healthy subjects [119].

Interestingly, two studies in asthma patients revealed the negative effect of airway cooling and nocturnal asthma, and the association between diurnal temperature variation above 10 °C and childhood asthma exacerbation [120,121]. These findings highlight the potential effect of environmental temperature changes on our immunity and may be one mechanism to explain the seasonal change of our immunity [122]. Nevertheless, it is important to note that these studies had investigated the environmental temperature variation and not the CRBT.

## 5. Conclusions

This review illustrates the potential implication of the CRBT in the immune response and the link between circadian disruption of the CRBT and diseases.

The CRBT is a marker of the central biological clock that results from metabolic heat production and is regulated by complex biological clock processes. Body temperature is easy to measure compared with other markers, such as cortisol or melatonin; however, central measurements seem more reliable in the study of circadian rhythms. The CRBT may be affected by many different factors, including environment, age, hormones, or treatment, and thus could be considered potentially involved in modifying immune functions in such contexts. It is now well-established that CRBT affects the expression of the molecular clock and in this way modifies the expression of all clock-controlled genes. Body temperature changes may positively affect immunity and can promote the activation, function, or traffic of immune cells; these changes are also associated with better resistance to infection in animal models. In humans, several pieces of evidence suggest that the disruption of CRBT is associated with many kinds of diseases and their severity, thus supporting the assumed association between CRBT and immune functions.

Most studies have focused on finding the presence or absence of a normal, 24-h circadian rhythm. However, recent studies using complex analysis suggest that circadian rhythm may change in all aspects (level, period, amplitude) and may be predictive of good or poor outcomes [99,105]. It would make sense that an adaptation of rhythms is physiological and necessary to trigger the immune response in response to stimuli. Therefore, it will be necessary for future studies to understand the different circadian disruption types by using appropriate models.

Finally, in the era of personalized medicine, and given the importance of circadian rhythm in the immune response, the monitoring of the CRBT appears to be an easy tool to predict outcomes and guide future studies in chronotherapy.

## Figures and Tables

**Figure 1 biology-10-00065-f001:**
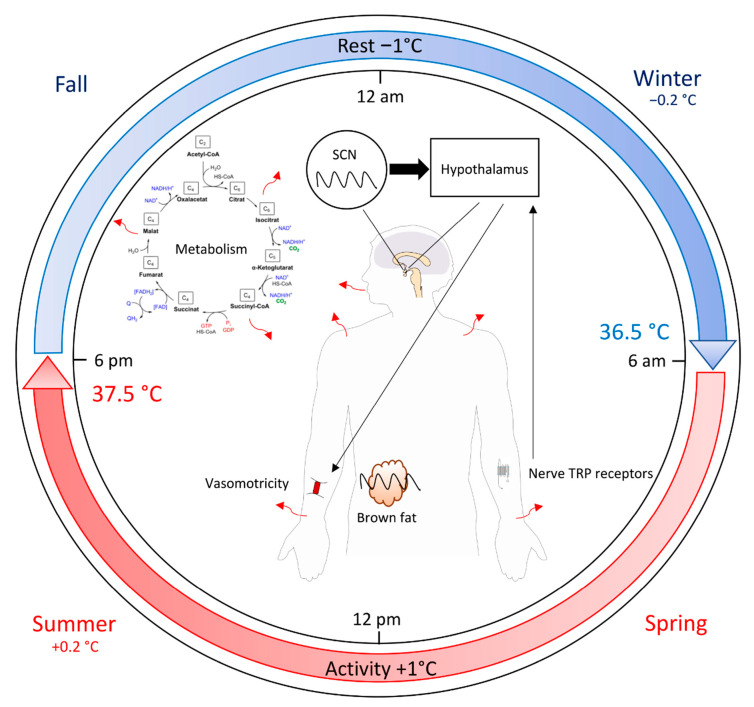
Origin, control, and variation of the body temperature in humans. The heat production is generated by the metabolism and the activity resulting in an increase of body temperature during the day. Heat loss is the consequence of sweating or vasomotor changes that regulate blood flow to the skin and mucous membranes via nerve sensors (thermal TRP) of the temperature and coordinated by the hypothalamus. Temperature oscillation is synchronized by the SCN but may also be generated by the brown fat driven by their cell molecular clocks. Red arrows represent the heat. The climate environment results in a change in body temperature around 0.2 °C between winter and summer. SCN—suprachiasmatic nucleus; TRP—transient receptor potential.

**Figure 2 biology-10-00065-f002:**
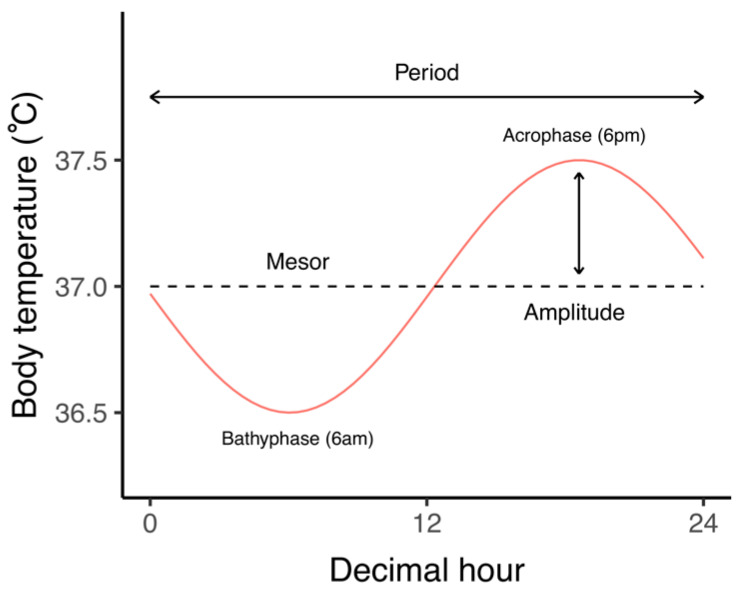
Circadian parameters of the body temperature. The graph represents the circadian rhythm of the body temperature (red curve). The dashed line represents the mesor (MESOR = Midline Estimating Scheme 24 h for a circadian rhythm). The amplitude is the difference between the mesor and the value at the peak. The acrophase is the time-of-day of the maximum value. The bathyphase is the time-of-day of the minimum value.

**Figure 3 biology-10-00065-f003:**
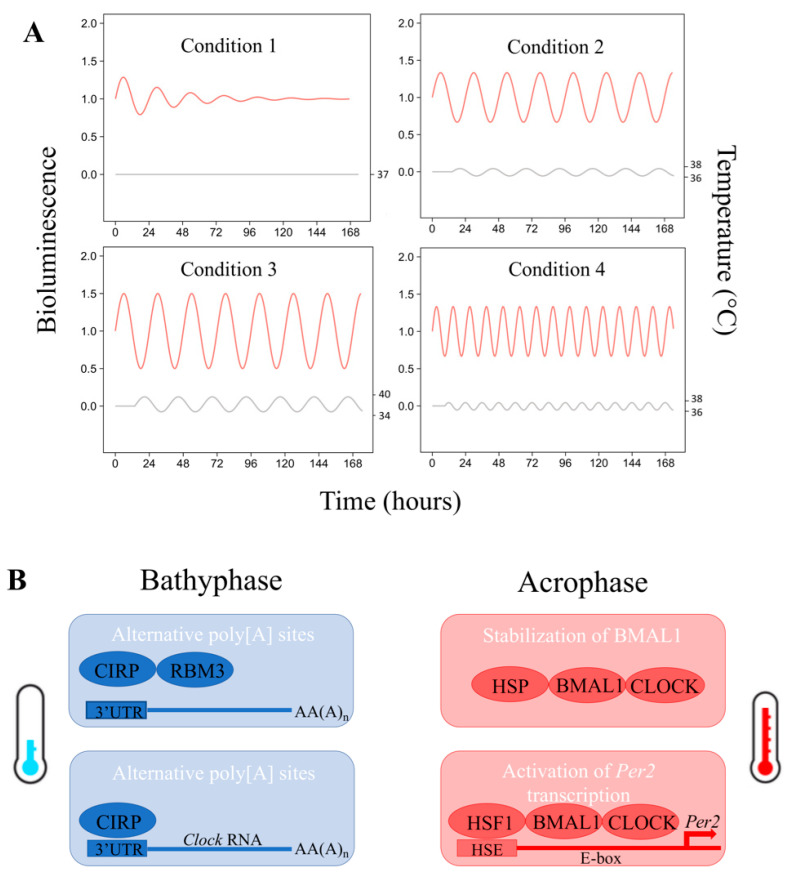
Molecular connection between the circadian rhythm of body temperature (CRBT) and the molecular clock. (**A**) Schematic representation of the effect of the temperature on the expression of the clock genes (e.g., Bmal1 in a model of cultured fibroblasts). Red curves represent the expression of Bmal1 quantified by bioluminescence. Grey curves represent the temperature cycles set in the incubator. From left to right and top to bottom, an example of cell culture maintained at 37 °C showing a progressive decrease of the oscillatory expression of the clock gene (Condition 1). Then followed by three examples of synchronization of the clock gene expression by three different conditions of temperature cycles: with an amplitude of 2 °C and a period of 24 h (Condition 2), an amplitude of 4 °C and a period of 24 h (Condition 3), and an amplitude of 2 °C and a period of 12 h (Condition 4) [11]. (**B**) Possible mechanisms involved in the temperature entrainment of molecular clocks in the peripheral tissues are depicted [36]. Cold temperature-induced RNA-binding proteins CIRP and RBM3 lengthen the 3′ untranslated region (3′-UTR) of target RNAs by suppressing their proximal polyadenylation sites. CIRP also promotes the nuclear export and translation of Clock RNA. On the other hand, warm body temperature transcriptionally activates *Per2* expression in an HSF1-dependent manner, whereas HSP90 post-translationally stabilizes BMAL1-CLOCK complex, a heterodimeric activator of the Per2 transcription.

**Table 1 biology-10-00065-t001:** Effect of the temperature level on the immune functions. This table lists the immune parameters that describe a circadian rhythm (~) under normal condition (normal CRBT) and changes (⬈: increase or ⬊: decrease) in immune features and functions in fever (or heat) and hypothermia. CHS—contact hypersensitivity; GVHD—graft-versus-host disease; KO—knock-out; WT—wild type.

	Trend	Immune System Variation	Species	Model	Ref.
**Fever**	⬈	DC migration	Human (healthy)	In vitro	[45]
	⬈	APC and T-cell interactions	Human (healthy)	In vitro	[45,46]
	⬈	TNF and IL-12 production	Mice (WT)	In vivo and in vitro	[36,46]
	⬈	IL-2 CD4^+^ T-cell production	Human (healthy)	In vitro	[47]
	⬈	IFN-γ CD8^+^ T-cell production	Mice (WT)	In vitro	[48]
	⬈	Ab-dependent complement-mediated lysis	Mice (healthy)	In vitro	[49]
**37.5 °C**	⬈	Lymphocyte adhesion and trafficking	Human/Mice (healthy)	In vitro	[50]
	~	Blood and tissue leukocyte number	Human (healthy)/Mice (WT/KO)	In vitro and in vivo	[51,52]
	~	Neutrophils ICAM-1 expression	Mice (WT/KO)	In vitro and in vivo	[52]
	~	Cytokine production	Human (arthritis)	In vivo and in vitro	[53]
	~	Phagocytic activity	Human (healthy)/Mice (WT/siRNA)	In vivo and in vitro	[54,55]
	~	Natural killer-cell activity	Human (healthy)	In vitro	[56]
	~	Whole-blood transcriptome	Human (healthy)	In vivo and in vitro	[41]
**36.5 °C**	⬊	DC tissue infiltration	Rat (WT)	In vivo and in vitro	[57]
**Hypothermia**	⬊	TNF and IL-12 production	Mice (WT with ischemic stroke)	In vivo and in vitro	[58]
	⬈	IL-10 production	Mice (WT with ischemic stroke)	In vivo and in vitro	[58]
	⬊	DC maturation	Mice (WT with tumor)	In vivo and in vitro	[59]
	⬊	T-cell-activating function	Mice (WT with tumor)	In vivo and in vitro	[59]
	⬊	IFN-γ T-cell production	Mice (endotoxemic)	In vivo and in vitro	[60]
	⬊	Lymphocyte tissue infiltration	Mice (WT with acute GVHD)	In vivo and in vitro	[61]
	⬊	Homing of lymphocytes	Mice (WT experiencing CHS)	In vivo and in vitro	[62]
	⬈	Neutrophil and macrophage infiltration	Rat (Muscle contusion injury)	In vivo and in vitro	[63]

## Data Availability

Not applicable.

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
