# Peer review of "A Tangled Threesome: Circadian Rhythm, Body Temperature Variations, and the Immune System"

_biology, 2021, doi:10.3390/biology10010065_

Round 1

Reviewer 1 Report

  1. The authors may want to focus on a relationship between two topics, not all three. CRBT is controlled by the circadian clock in mammals. However, mammals have homeostatic temperature-mediating ability that may not necessarily have anything to do with the circadian clock, and this seems like a topic too distant to be easily transitioned within the paper, since the paper has significant portions that only pertain to the temperature mediation unrelated to the circadian clock. The immune system is another disparate topic, and it seems that direct evidence is lacking regarding the disruption of circadian clock leading to diminished immune function, especially in a clinical setting.
  2. Isn't temperature mediation a simple homeostatic process that is relatively independent from the circadian clock? If so many factors (line 38-45) affect the body temperature, how can you tell it apart that the circadian clock is really affecting CRBT? More biochemical correlation is expected to draw a clear connection between the circadian clock and CRBT.
  3. There is no point in talking about aging and less efficient temperature-controlling ability when it may have nothing to do with the circadian clock itself? The same goes for gender. Also, the part on temperature's effect on immune function in other domains of life does not go well with the rest of the paper. This is because the paper mostly talks about the mammalian system, especially with the clinical part. It would be better to just focus on the mammalian system, or the paper would be too broad.
  4. Temperature variation from the environment as the input and CRBT as the output are topics that are still too disparate. If temperature can control the circadian clock as an entrainment cue and the affected CRBT produces a rhythmic temperature variation, wouldn't this mechanism feed back into the system and collapse? In that sense, it's still not clear to me if having a fever is the body's intentional immune response in the event of a disease or a result of having a disease, unintentionally causing circadian clock and CRBT disruption.

Author Response

1. The authors may want to focus on a relationship between two topics, not all three. CRBT is controlled by the circadian clock in mammals. However, mammals have homeostatic temperature-mediating ability that may not necessarily have anything to do with the circadian clock, and this seems like a topic too distant to be easily transitioned within the paper, since the paper has significant portions that only pertain to the temperature mediation unrelated to the circadian clock. The immune system is another disparate topic, and it seems that direct evidence is lacking regarding the disruption of circadian clock leading to diminished immune function, especially in a clinical setting.

Response: The effect of the CRBT (in the sense, that the temperature oscillation would be a homeostatic condition with specific ability, different and independent from the temperature level) on the immune effectors has never been directly assessed, neither in humans nor in animals or in vitro. It could have a specific role, or not? We still don’t know…

One hypothesis would be that an increased temperature prepares the organism to fight aggression specifically in the daytime (it is more probable that trauma or infection would occur during the day rather than the night). The level of the body temperature increases in the daytime (with a peak at 6:00 pm) probably to activate immune functions (increase immune cells migration or delivery, cytokines productions, etc.) as described in section 3.2 of our review, and may prepare the organism against aggression. This observation has also been made for the number of circulating leukocytes that are at a higher level during the daytime probably also to prepare the organism against aggression, as hypothesized by the authors [43]. The level of the temperature is lower during the night, maybe to regenerate the immune function? It has been demonstrated that an infection is more severe when mice are infected at the resting phase than during the active phase (Gibbs J, et al. An epithelial circadian clock controls pulmonary inflammation and glucocorticoid action. Nat Med 2014;20(8):919-26).

Thus, to support this hypothesis, it appears important to discuss the effect of the temperature level (high, like fever, or low, like hypothermia) on the immune function and in disease, which is well documented, rather than only the potential effect of an independent ability of the circadian variation of the temperature which has been poorly studied or only in plants or ectodermic animals. The CRBT is so maybe a mechanism to mobilize the immunity during daytime and restore the system during the night.

To increase comprehensibility, and better express that point we mentioned this notion in section 3.2 lines 253-258.

Concerning the immune system, indeed direct evidence is lacking regarding the disruption of the circadian clock leading to diminished immune function, however, there are dozens of clinical studies demonstrating the link between circadian disruption and disease severity (and by symmetry, there is a lower circadian disruption in less severe disease) suggesting a link between circadian disruption and impaired immune function as detailed in section 4 for CRBT. In animals, KO models for clock genes demonstrated more severe disease too.

2. Isn't temperature mediation a simple homeostatic process that is relatively independent from the circadian clock? If so many factors (line 38-45) affect the body temperature, how can you tell it apart that the circadian clock is really affecting CRBT? More biochemical correlation is expected to draw a clear connection between the circadian clock and CRBT.

Response: It is commonly accepted that the biological clock and mainly the supra-chiasmatic nucleus probably control or influence the circadian rhythm of the body temperature (this is discussed in section 2.1: studies in squirrels with complete ablation of the SCN and the clock control of the brown adipose tissue are the two main proofs). But, direct biological evidence is indeed lacking. One can consider that increase in temperature may be only due to an increase in metabolism and heat production. One study in a mouse model demonstrated that the clock gene Per2 is essential in the daily maintenance of animal locomotor activity and body temperature rhythmicity [39]. But, once again, a disturbance of locomotor activity can lead to a disturbance in the rhythmicity of body temperature.

However, it doesn’t change the pertinence of our review with the main objective to discuss the potential influence of temperature variability on the control of the immune system, whatever the control or production of the temperature variability.

3. There is no point in talking about aging and less efficient temperature-controlling ability when it may have nothing to do with the circadian clock itself? The same goes for gender. Also, the part on temperature's effect on immune function in other domains of life does not go well with the rest of the paper. This is because the paper mostly talks about the mammalian system, especially with the clinical part. It would be better to just focus on the mammalian system, or the paper would be too broad.

Response: In section 2.3, the objective was to describe physiological modulation of the CRBT and the potential link with different profiles of immune response mainly according to the season, age, and sex. To better point out this notion, we rephrased the introduction of this section (lines 117 to 121).

The main objective of section 3. is to discuss the biological effect of temperature variations on the immune function in biology. As discussed above, there is no direct evidence in mammals. However, there is some evidence in plants and ectodermic animals of an improvement of immune function in the context of temperature variation compared to a constant temperature. Thus, due to the lack of evidence in mammals, it appears important for us to describe studies in other domains of life than mammals that reinforce our hypothesis of control of the immune system by the CRBT.

To less restrain the manuscript on the mammalian system we deleted the notion of CRBT and rephrased the title of section 3. and its sub-sections (lines 161, 162, 307, 332).

4. Temperature variation from the environment as the input and CRBT as the output are topics that are still too disparate. If temperature can control the circadian clock as an entrainment cue and the affected CRBT produces a rhythmic temperature variation, wouldn't this mechanism feed back into the system and collapse? In that sense, it's still not clear to me if having a fever is the body's intentional immune response in the event of a disease or a result of having a disease, unintentionally causing circadian clock and CRBT disruption.

Response: We would suggest the reverse. Temperature can control the circadian (molecular) clock as an entrainment cue, so, the clock produces the rhythmic temperature variation which in turn can drive the clock itself. That would be a self-autonomous cycle?

Our main objective is to discuss the impact of the temperature variation on the immune system in biology in general, whether temperature variation from the environment on plants or ectodermic animals or CRBT on the immune system in mammals. One way to explain the effect of a temperature variation on the immune system would be the impact of the temperature variation on the genome expression and thus on the expression of the genome involved in the immune pathways. The fact that the CRBT is entrained by the biological clock (central system) is another topic, indeed, and probably that the molecular clock is not directly involved in generating the CRBT.

From our point of view, having a fever (or increase the amplitude of the temperature, or change the rhythm of the temperature) is an intentional immune response (involving, among others, the production of TNF by the immune system) 1) to activate/mobilize the immune parameters (as described in section 3.2 of our review), and 2) to trigger, amplify, synchronize the expression of circadian genes and/or clock-controlled genes and activate immune or metabolic pathways involved in the immune response or related to the aggression.

We did not discuss in this review how the temperature may change in the context of infection or disease because we would like to mainly focus on the effect of temperature variations on the immune system and not the reverse.

Extra: In addition, reference 11 was doubled with reference 33 in our previous version. We deleted reference 33 and reordered all references.

Reviewer 2 Report

Most of my comments have been addressed well. I have a few minor comments:

- I had suggested to include in Fig. 4 the information whether the presented studies were performed in clinical populations/animal models of disease or healthy subjects. The authors replied that details on the species were already specified. However, I was not referring to the species here, which, as the authors correctly point out, is already specified in the figure. I was referring to the distinction whether the studies were performed in clinical populations versus healthy humans (or in animal models of disease versus genetically unmodified animals in case of animal studies). The authors may consider to add this information.

- Lines 417f: To improve readability, I would write something like “… thus supporting the assumed association between CRBT and immune functions.” or similar.

- Lines 364f: What does it mean that an increase “in temperature amplitude … was associated with non-septic patients”? Were septic patients characterized by the absence of an increase in temperature amplitude, whereas non-septic patients did show such an increase? The authors might consider rephrasing this sentence to improve comprehensibility.

Author Response

Most of my comments have been addressed well. I have a few minor comments:

- I had suggested to include in Fig. 4 the information whether the presented studies were performed in clinical populations/animal models of disease or healthy subjects. The authors replied that details on the species were already specified. However, I was not referring to the species here, which, as the authors correctly point out, is already specified in the figure. I was referring to the distinction whether the studies were performed in clinical populations versus healthy humans (or in animal models of disease versus genetically unmodified animals in case of animal studies). The authors may consider to add this information.

Response: Lines 270-274: As suggested, we added the precision of whether studies were performed in a healthy or disease context.

- Lines 417f: To improve readability, I would write something like “… thus supporting the assumed association between CRBT and immune functions.” or similar.

Response: Lines 463-464: As proposed, we changed the sentence for the proposed one as follows: “…thus supporting the assumed association between CRBT and immune functions.”

- Lines 364f: What does it mean that an increase “in temperature amplitude … was associated with non-septic patients”? Were septic patients characterized by the absence of an increase in temperature amplitude, whereas non-septic patients did show such an increase? The authors might consider rephrasing this sentence to improve comprehensibility.

Response: Lines 406-408: the sentence was indeed misleading. No, it means that temperature amplitude in non-septic patients was higher than in septic patients, but, because the amplitude was lower than the normal range in septic patients… So, we rephrase as follows: “… authors found a decrease in temperature amplitude below normal range at ICU admission in septic patients, and lower severity of disease and better clinical outcomes in patients with higher temperature amplitude”

Extra: Besides, reference 11 was doubled with reference 33 in our previous version. We deleted reference 33 and reordered all references.

Round 2

Reviewer 1 Report

All the points I have raised have been addressed satisfactorily.

This manuscript is a resubmission of an earlier submission. The following is a list of the peer review reports and author responses from that submission.

Round 1

Reviewer 1 Report

1.       Although the paper tries to suggest the relationship between temperature, CRBT, and immune system, they are topics too broad and require intense transitions within the paper. For example, the entrainment using temperature variance (input) and the clock-controlled temperature modulation by the body (output) are completely different topics, yet it seems that the topics are scattered throughout the paper. In addition, how other organisms respond and react to temperature changes and how they modulate their own body heat are often not relevant to the main point of the paper, especially when there is no transition between each of them. Personally, “4. Circadian disruption of the temperature and diseases” was really informing and coherent and perhaps the authors could elaborate more on this part.

2.       SCN controls a lot of temporal expressions besides temperature. What makes you so sure that CRBT activity is directly controlled by SCN and this controls the immune system up to a meaningful level? Supposedly, the circadian clock can support the immune system without going through CRBT, right? I want to hear more about CRBT’s unique effect on the immune system.

3.       The cause and effect of having a fever or hypothermia in relation to CRBT is unclear. Does having an immune response exert an intentional or unintentional effect on the clock? What about the clock’s role in controlling the body temperature in the event of an infection?

4.       Authors may elaborate more on some beneficial aspect of having a CRBT and its positive implications on the immune system.

Author Response

Dear Reviewer,

We are very grateful for your careful review. Due to the COVID19 crisis, we were not able to submit a revised version of our review before. We are sincerely sorry for this delay.

Reviewer 2 Report

Coiffard et al. describe in their review the physiological basis and molecular regulation of the prominent circadian rhythm in body temperature, followed by an overview of the literature showing causal or correlational associations of temperature changes with immune parameters and related diseases.

The review deals with a very interesting and somewhat underappreciated topic that is definitely of high clinical value. My main concern is that data directly investigating the role of the circadian rhythm of body temperature (CRBT) on immune parameters is scarce and, thus, a review article focusing on this aspect might be a bit premature. This concern could be addressed by adapting the main title and some subtitles of the article, reducing the emphasis on circadian rhythms in body temperature. A broader title focusing more on general effects of temperature on immune parameters and related diseases would in my opinion be more appropriate. I don’t see much “new insight” in the specific role of the CRBT. Alternatively, the article might be written as an opinion article summarizing (mostly indirect) evidence on the role of the CRBT on immune parameters. See below for specific comments.

Major points

- The English language needs some editing.

- The title of the review is somewhat misleading, as effects of the CRBT on immune parameters are not very well investigated and the review mainly summarizes effects of temperature deviations from normal on immune parameters. Also, because the review also includes effects of temperature that are not always directly related to immune functions, a broader title might me more appropriate.

- The title of the section 2.3. is in my opinion not appropriate. The section describes associations between several factors, such as season, age and sex, on temperature rhythms, but the rhythms are not necessarily “disrupted”.

- Section 2.4 overlaps with sections 4.3. and 4.4 and should be restructured.

- Lines 196f: “Thus, temperature changes (like fever) most likely participate in the activation of the peripheral clocks in order to prepare a systemic immune reaction.”: The reference to the immune system is not clear here, because the role of fever in immune functions is discussed only later.

- The title of Figure 3 mentions immune pathways, but no such pathways are shown in the figure.

Section 3.2.: This section mainly describes effects of unphysiological temperature, like fever and hypothermia, rather than effects of the CRBT on the immune system. This should be reflected in the title of the section.

- Lines 241ff: “Thus, an elevation of body temperature above 38°C, commonly called fever, is likely a mechanism to prepare the immune system to fight against aggression, such as infection, but could also be a mechanism of clock activation and an increase of temperature amplitude a trigger of the molecular clock expression and immune function activation.”: This sentence is a bit complicated to read and seems partly redundant. I would suggest to separate it into two sentences.

- The studies in references 58-76 investigated effects of high/low temperature on immune parameters or circadian rhythms in immune parameter, but they don’t refer to effects of the CRBT on immune parameters. Figure 4 and its title are therefore somewhat misleading. I would suggest to indicate more details of the different studies in the figure or the text, including also information on whether the experiments were performed in vivo or in vitro and whether they were performed in clinical populations/animal models of disease or healthy subjects. Furthermore, I would separate studies directly investigating associations of temperature with immune parameters and those investigating circadian rhythms in immune parameters (independently of temperature). Although the idea that circadian rhythms in immune parameters are mediated by circadian changes in body temperature is interesting and convincing, it is mainly a hypothesis and this has to be made clearer in the manuscript.

- Why are references 78 and 80 not included in Figure 4?

- The authors describe effects of environmental temperature in plants and ectotherm animals. Although these findings are interesting, I am not sure whether they can be translated to endotherm animals. This should be discussed.

- Please indicate whether the temperature cycles mentioned in lines 265 and 270 were 24-hour long cycles.

- Lines 299ff: “Although exposed to environmental changes, such as nutrient availability [94], and the level of host-derived immunity [93], the CRBT may thus be an important circadian parameter influencing the gut microbiome.”: Again, the putative role of the CRBT is purely speculative here, which should be made clear.

Lines 374f: “In animals, some studies found an association between disruption of the CRBT and infectious disease …”: Please specify the results of the mentioned studies.

- Lines 394-398: These findings relate to changes in environmental temperature and not to the circadian rhythm of the body temperature, which should be more clearly distinguished.

- The following statements in the conclusion are in my view too speculative and not fully supported due to the lack of literature:

- Line 400: “This review illustrates the implication of the CRBT in the immune response and the link between circadian disruption of the CRBT and diseases.”

- Lines 405ff: “The CRBT may be disrupted by many different factors including environment, age, hormones, or treatment, and thus impacts functions under the control of the clock.”

- Lines 410f: “In humans, the disruption of CRBT is widely associated with all kind of diseases and their severity highlighting the association between circadian rhythm and the immune response.”

Minor points

Lines 114ff: “This section aims to highlight the potential association between physiological change in CRBT and the effect of some factors, i.e., seasons, age, and sex, also well-known associated with different immune response profiles and infectious risks [21,22].”: A word is missing in this sentence.

- Lines 131f: “An association between the phase of the CRBT and sleep–wake cycle was made in elderly subjects [27].”: I’m not sure how this sentence connects to the sentence before and afterwards.

- Lines 238ff: “High body temperatures in the range of physiologic fever in mammals (~38–41°C) have multiple effects on mammalian antigen-presenting cells that promote their T-cell-activating function and the adaptive immune response [78].”: It would be interesting to know which are the “multiple effects” of high body temperatures mentioned in this sentence.

- Line 270: Please indicate the direction of the mentioned difference in resistance (increase or decrease).

- Lines 292f: What do the authors mean with “temperature-compensated circadian rhythm”?

- Lines 304ff: “… experience significantly different phenoloxidase activity…”: Please replace the word “different” and indicate the direction of the effect.

- How was the “temperature rhythmicity” mentioned in line 369 assessed?

- Please add the references to the “recent studies” mentioned in line 414.

Author Response

Dear Reviewer,

We are very grateful for your careful review. Due to the COVID19 crisis, we were not able to submit a revised version of our review before. We are sincerely sorry for this delay.

First Author
